# Metabolic Dynamics of In Vitro CD8+ T Cell Activation

**DOI:** 10.3390/metabo11010012

**Published:** 2020-12-28

**Authors:** Joy Edwards-Hicks, Michael Mitterer, Erika L. Pearce, Joerg M. Buescher

**Affiliations:** 1Department of Immunometabolism, Max Planck Institute of Immunobiology and Epigenetics, Stübeweg 51, 79018 Freiburg im Breisgau, Germany; edwards-hicks@ie-freiburg.mpg.de (J.E.-H.); pearce@ie-freiburg.mpg.de (E.L.P.); 2Metabolomics Core Facility, Max Planck Institute of Immunobiology and Epigenetics, Stübeweg 51, 79018 Freiburg im Breisgau, Germany; mitterer@ie-freiburg.mpg.de

**Keywords:** T cell, activation, metabolic reprogramming, metabolomics, lipidomics, LC-MS, FIA-MS, time course

## Abstract

CD8+ T cells detect and kill infected or cancerous cells. When activated from their naïve state, T cells undergo a complex transition, including major metabolic reprogramming. Detailed resolution of metabolic dynamics is needed to advance the field of immunometabolism. Here, we outline methodologies that when utilized in parallel achieve broad coverage of the metabolome. Specifically, we used a combination of 2 flow injection analysis (FIA) and 3 liquid chromatography (LC) methods in combination with positive and negative mode high-resolution mass spectrometry (MS) to study the transition from naïve to effector T cells with fine-grained time resolution. Depending on the method, between 54% and 98% of measured metabolic features change in a time-dependent manner, with the major changes in both polar metabolites and lipids occurring in the first 48 h. The statistical analysis highlighted the remodeling of the polyamine biosynthesis pathway, with marked differences in the dynamics of precursors, intermediates, and cofactors. Moreover, phosphatidylcholines, the major class of membrane lipids, underwent a drastic shift in acyl chain composition with polyunsaturated species decreasing from 60% to 25% of the total pool and specifically depleting species containing a 20:4 fatty acid. We hope that this data set with a total of over 11,000 features recorded with multiple MS methodologies for 9 time points will be a useful resource for future work.

## 1. Introduction

CD8+ T cells are critical effectors of the adaptive immune defense against pathogens and tumors. In response to an immune challenge, naïve CD8+ T cells (TN) are activated to form cytotoxic effector T cells (TE), which undergo clonal expansion and synthesize effector molecules to kill infected or malignant cells [1]. Rapid metabolic reprogramming during T cell activation is required to support the increased biosynthetic demands of epigenetic remodeling, proliferation, and cytokine production [2]. The availability of extracellular metabolites, including glucose, amino acids, and lipids affects TE metabolic reprogramming, and anti-tumor immunity is restricted by nutrient competition and metabolic byproducts in the tumor microenvironment [3,4,5]. Metabolites are also key mediators of early T cell signaling events at the plasma membrane, integrating extracellular signals with signal transduction networks that promote activation [6].

Cells contain a vast range of small molecules such as inorganic ions, water-soluble metabolites, and lipids. In addition, the cellular concentration of these molecules can differ by multiple orders of magnitude [7,8]. While no single analytical method can simultaneously capture the whole range of small molecules, state-of-the-art metabolomics methods can deliver over 1000 features and over 100 annotated metabolites. Generally, chromatography methods differ in the spectrum of metabolites that they cover (Table 1).

Every one of the employed analytical methods has limitations that must be taken into account during sample preparation and data analysis. The retention of metabolites and their chromatographic separation can be negatively impacted by contamination in the sample. For the LC-QTOF HILIC method, these are typically residual pH buffers or acids from the culture medium or buffers that were used to wash the culture medium off the cells. For the LC-QTOF polar RP method, residual organic solvents such as methanol that was used to extract metabolites from cells can be problematic. For the LC-QTOF Lipids method, strong organic solvents such as chloroform or methyl-tert-butyl ester (MTBE), which are often used for lipid extraction, can negatively impact the chromatography.

The settings of the mass spectrometer will influence which metabolites can be detected and how the recorded features can be annotated. In the context of this study, a feature is defined by mass and, for LC-MS data, retention time. The signal intensities of a feature can be compared across multiple samples and used in statistical analyses. Likewise, we use the term “metabolites” to denote compounds with a known structure and biological function. By matching measured information of features (such as mass, isotope pattern, fragment spectrum, or retention time) to known properties of metabolites, the features can be annotated with metabolite names or sum formulas. The more properties of a feature can be matched to a metabolite, the higher the confidence in the resulting annotation.

Negative ionization polarity favors the detection of metabolites containing carboxylic acid, phosphate, or sulfate moieties and is also useful for sugars. Positive ionization polarity favors metabolites containing amine moieties. In our experience, a compound containing conjugated pi-electron systems such as aromatic compounds also ionize reasonably well in positive mode. One-dimensional mass spectrometry (MS1) on a high-resolution mass spectrometer allows the annotation of the feature with sum formulas using the measured exact mass and isotope pattern. In addition, this information can be used together with known retention times to annotate known metabolites among the features. In two-dimensional mass spectrometry (MS2), ions recorded in an MS1 scan are selected, isolated by a quadrupole, fragmented, and the mass spectrum of the fragments is recorded. This fragment spectrum can be matched against spectral libraries to annotate metabolites and it can be used to improve the sum formula annotation of the corresponding MS1 feature.

As the complex interplay between metabolism and T cell function is unraveled, there is a growing interest among immunologists in measuring dynamic metabolic changes in specific cell states. In this manuscript, we carry out a longitudinal analysis of primary CD8+ T cell activation in vitro, taking time points every 12 h for 4 days in total (9 time points) to capture changes in the metabolome with fine-grained time resolution. Furthermore, we achieve broad coverage of metabolites across this time course using combined methodologies: positive and negative ionization modes, 2 flow injection analysis (FIA)-MS, and 3 different LC-MS/MS methodologies.

## 2. Results

To determine metabolic dynamics of CD8+ T cell activation, we activated CD8+ T cells in vitro and extracted intracellular and extracellular metabolites every 12 h for 96 h in total. Primary murine CD8+ T cells were isolated from spleens of 8-week-old C57BL/6J mice and activated in vitro under standard conditions (see Material and Methods for details). Before each sampling, the cell density in the culture was determined and the volume that contained 3 × 10^6^ cells was sampled. After extraction, metabolite extracts were split into aliquots for all analytical methods. To enable the observation of monotonous and transient time courses in metabolites during the activation from TN to TE, we chose to sample every 12 h for 4 days, yielding a total of 9 time points (Figure 1). In addition, a sterile well (with the medium but without cells) control sample (blank) was taken at the initial time point.

The medium that the cells were cultivated in contains many of the same metabolites that can also be found inside the cells. Therefore, we washed the cells prior to extraction to minimize medium contaminations. However, washing can impact the metabolic composition of the cells [9]; therefore, we only washed once. In our experience, additional washing steps can further reduce contaminants at the cost of losing high-energy metabolites such as adenosine triphosphate (ATP) and fructose bisphosphate (FBP). About half of the features were found to be higher in cell extracts than in a sterile well control with a *p*-value smaller than 0.05 in a pair-wise t-test. For all subsequent analyses, we only used features that were above the blank for at least one time point.

### 2.1. Polar Metabolites

To quickly obtain an initial overview of the metabolic dynamics of CD8+ T cell activation without chromatographic bias, we performed FIA in both positive and negative ionization mode. Taken together, the FIA measurements yielded over 2000 metabolic features above the blank, 60% of which showed a significant time trend in a likelihood ratio test (Table 2). This observation indicates that CD8+ T cell activation not only affects a few metabolic pathways but is also accompanied by global metabolic changes. For a more robust identification of changing features, we performed partial least squares (PLS) analysis with sampling time as the predicted variable on the combined positive and negative ionization mode FIA data (Figure 2). Both the 1st and the 2nd PLS dimensions help in separating the groups of replicates along the time trajectory. Notably, the separation is greater for earlier time points, indicating that the major metabolic differences occur in the first 48 h of CD8+ T cell activation.

To identify the areas of metabolism that are most affected during the activation of CD8+ T cells, we used the sum of the absolute loading values from the PLS analysis and the KEGG (Kyoto Encyclopedia of Genes and Genomes) definition of metabolic pathways [10] as input for a gene set enrichment analysis (GSEA) (Figure 3). As we have used sampling time as a predicted variable in our PLS analysis, the absolute values of the loadings can be interpreted as a measure of how much each feature contributes to the overall time-dependent changes. In case multiple metabolites in the database have the same sum formula or very similar mass, one measured FIA feature can be annotated with multiple metabolites (see methods for details). For the GSEA analysis, we then assigned the measured data to every one of these metabolites. Notably, central carbon metabolism, tricarboxylic acid (TCA) cycle, and amino acid metabolism are among the top-scoring pathways. This is in line with previous findings that found similar pathways enriched in metabolic differences by ^13^C tracing [11], even though the initial annotation of FIA features is based on exact mass only.

Two of the major metabolic effects during the activation of CD8+ T cells that have been described previously are the increase in glycolytic lactate production and the decrease in fatty acid oxidation (FAO) in activated cells compared to naïve CD8+ T cells [11,12]. In line with these reports, we find the concentration of lactic acid to increase and that of carnitine to decrease (Figure 4). This confirms that our results are comparable to other studies with in vitro activated CD8+ T cells. Interestingly, the decrease in carnitine seems to happen earlier than the increase in lactic acid, indicating that the activity of glycolysis and FAO are not directly coupled.

To obtain data with higher confidence in the annotation and quantification of features than the ultra-high throughput FIA technology can provide, we measured aliquots of all time points by LC-MS. To cover the metabolic pathways that were identified as relevant based on the FIA data, we used 2 chromatographic methods that are suitable for a broad range of polar metabolites including glycolysis, TCA cycle, and amino acids (Table 1). In addition, we use a chromatographic method suitable for lipids to complete the picture even though we only had FIA data from the aqueous fraction of the cell extracts because our FIA pipeline has been developed and tested only for polar metabolites. Measurements using a QTOF-MS enable a data-driven discovery metabolomics approach that allows for the identification of unexpected metabolic dynamics. We provide the raw mass spec data (www.ebi.ac.uk/metabolights/MTBLS2145) in addition to our processing of the data sets (Appendix A) as a resource to allow the community to further explore the data.

For an initial overview of the polar metabolome measured by LC-QTOF-MS, we performed a principal component analysis on the combined HILIC and reversed-phase data set (Figure 5). Principal component 1 (PC1) represents more than one-third of the total variance in the data set and separates the replicate measurements in the first half of the experiment better than those from the second half of the experiment. This indicates that the major metabolic changes occur in the first 48 h following activation of the T cells, thus confirming the above observation that was based on the FIA data. In a plot of PC3 vs. PC4, groups of replicates are still reasonably well separated (Appendix A) indicating that more subtle and complex metabolic dynamics occur in addition to the major changes.

In line with the GSEA analysis of the FIA data, there are many amino acids and related metabolites as well as nucleotides and related metabolites among the annotated features that contribute the most to PC1 (Appendix A). Interestingly, the two polyamines spermidine and spermine are also within the top 20 in this ranking.

Recently, the importance of polyamines was described for several immune cell types. Polyamines were described to accumulate in NK cells and in CD4+ T cells following activation from naïve state [13,14] and this accumulation was shown to be required for appropriate effector function. In IL4 activated macrophages, blocking of polyamine biosynthesis blunted their activation [15]. In parallel to these findings, we observe the two polyamines spermidine and spermine to also accumulate in CD8+ T cells following activation (Figure 6). Interestingly, spermidine levels sharply increase within the first 24 h of in vitro activation and plateau after 36 h. Spermine, which is produced from spermidine in polyamine biosynthesis (Figure 7), follows a similar time profile as its precursor, but with a few hours delay, reaching a plateau at around 48 h. S-adenosyl-methionine (SAM), which is an essential cofactor in polyamine biosynthesis, parallels the sharp increase in spermidine. Notably, the byproduct methyl-thioadenosine (MTA) increases only transiently and coincides with the period of greatest polyamine production. Precursors further upstream of spermidine remain largely unchanged (arginine, ornithine) or were not detected (putrescine).

### 2.2. Lipids

For a global view of the dynamics in the lipid fraction, we performed a principal component analysis of the combined positive and negative mode lipid data set. Even more pronounced than in the polar metabolome, the major changes in the lipid composition occur in the first 48 h of the experiment (Figure 8). The dominant changes are the increasing levels of the major membrane lipid phosphatidylcholine (PC), the major storage lipid triacyl glycerol (TAG), and hexosylceramide (HexCer) (Figure 9). Minor membrane lipids such as phosphatidylethanolamines (PE) and the lyso forms of PC and PE (LPC and LPE) increased transiently between 24 and 72 h post-activation before returning to values similar to pre-activation. Sphingomyelin (SM) decreased slightly over time. The increase observed for PC and other membrane lipids cause the total amount of lipids to increase over time. This is likely to be a direct consequence of the increase in cell size after activation (Appendix A).

For PC, the major class of membrane lipids, we asked if all observed species increased abundance or if there were differences in the time trend of the lipid species within this class that pointed toward a change in membrane composition. Following activation, CD8+ T cells accumulated PCs with fewer double bonds (predominantly 1–2 double bonds) and depleted polyunsaturated PCs (4 or more double bonds) (Figure 10, Appendix A). Overall, this resulted in decreased polyunsaturated PC species during T cell activation from ~60% in TN to ~25% in 48h activated TE. PCs generally contribute to >50% of plasma membrane lipid, and as saturated lipids are described to promote the formation of ordered domains that contain active signaling complexes [16], this shift in PC acyl chain composition could have important consequences for TE signaling. A recent study showed oncogenic epidermal growth factor signaling was dependent on membrane phospholipid remodeling to increase the abundance of saturated PCs [17]. Membrane lipid composition could similarly impact T cell signaling and is an interesting area for further research.

Previously, it has been reported that polyunsaturated membrane lipids can cause ferroptosis in cancer cells that travel in the bloodstream, and this can be reduced in cancer cells that first travel to the lymph where they accumulate monounsaturated membrane lipids [18,19]. It is, therefore, tempting to speculate that the observed shift toward less unsaturated PCs helps to prepare for the release of effector CD8+ T cells from the spleen into the bloodstream. Moreover, it has been found that increased order of membrane lipids protects cytotoxic T cells against perforin, one of the agents that they use to kill their target cells [20]. This protection was reported to be dependent on the order of the acyl chains and not the head group [21]. The shift toward fewer polyunsaturated membrane lipids might contribute to this protective effect.

We next asked if the trend toward the reduced abundance of PCs with more polyunsaturated fatty acids could be attributed to the occurrence of any particular fatty acid. To this end, we analyzed the lipid data recorded by LC-QTOF-MS in negative ionization mode specifically looking for features that represent formic acid adducts of PCs (Figure 11). This revealed that, in particular, PCs containing FA 20:4 (likely arachidonic acid) decreased sharply in abundance within the first 48 h of the experiment. Arachidonic acid is the precursor for the synthesis of eicosanoids, an important class of signaling molecules with effects on T cells [22]. Some eicosanoids can also be produced by T cells and they have been attributed roles in chemotaxis and adhesion to epithelial cells [23]. For a T cell line and for mast cells it has been shown that membrane phospholipids serve as a reservoir for arachidonic acid that can be mobilized when this fatty acid is needed as a precursor for eicosanoids [24,25]. These findings suggest that the shift in PC composition observed during CD8+ T cell activation could be the result of early signaling events.

## 3. Discussion

Detailed resolution of metabolic dynamics is needed to advance the field of immunometabolism. Here, we outline methodologies that when utilized in parallel achieve broad coverage of the metabolome. By coupling multiple LC-MS methods with fine-grained time course measurements, we find diverse metabolic changes during CD8+ T cell activation. Because our data was generated from aliquots of the same samples, we can meaningfully compare the timing of metabolic changes across classes of metabolites. For example, the timing of the depletion of phospholipids containing FA 20:4 matches the accumulation of the polyamines spermidine and spermine, which happen in the first 36 h following activation and plateau from 48 h onwards. In addition, the fine-grained time course allows us to detect transient metabolite changes such as the accumulation of methyl-thioadenosine at the 36 h and 48 h time points.

In this study, we highlight selected metabolites and lipids because of their biological relevance. Future systematic analysis of the data will likely yield additional insights and generate novel hypotheses. Therefore, we provide processed and raw data as a resource for immunologists to further investigate metabolic pathways of interest, and as a starting point for future research. We expect that the development of new statistical methods that exploit the time course relationship among the data points will support these efforts.

## 4. Materials and Methods

### 4.1. Mouse Lines

C57BL/6J mice were purchased from The Jackson Laboratory (JAX 000664). All mice were maintained at the Max Planck Institute of Immunobiology and Epigenetics and cared for according to the Institutional Animal Use and Care Guidelines.

### 4.2. Primary T Cell Cultures

Spleens harvested from 9, 8-week-old C57BL/6J mice were mashed in PBS and filtered through a 70 µm strainer. Red blood cells were lysed and CD8+ cells were isolated using the EasySep CD8+ T-cell isolation kit (Stem Cell Technologies, no. 19753) according to the manufacturer’s protocol. From 9 mice in total, CD8+ T cells from 3 mice were pooled to generate 3 biological replicates with sufficient T cells for the full time course. CD8+ T cells were activated at 1.5 × 10^6^/mL for 48 h using plate-bound anti-CD3 (5 μg mL^−1^; InVivoMAb anti-mouse CD3, BioXCell no. BE0002) and soluble anti-CD28 (0.5 μg/mL; InVivoMAb anti-mouse CD28, BioXcell no. BE0015) in RPMI 1640 medium (Invitrogen) supplemented with 10% fetal calf serum (Gibco), 4 mM L-glutamine, 1% penicillin–streptomycin, hrIL-2 (human recombinant interleukin 2, 100 U ml^−1^, Peprotech), and 55 μM beta-mercaptoethanol at 37 °C in a humidified incubator containing atmospheric oxygen supplemented with 5% CO_2_. Cells to be harvested at different time points were plated in separate wells of a 6-well plate to avoid continuously disturbing cultures during activation. Every 12 h, cells in the relevant well were suspended in their culture medium by pipetting and counted using a Neubauer chamber. A volume equivalent to 3 × 10^6^ cells was removed for sample prep (see below), and the remaining cells in that well were discarded. Following 48 h activation, every 24 h T cells were counted and plated back at 1.5 × 10^6^ in RPMI 1640 medium as above (minus CD3 and CD28) to prevent cell density from getting too high due to increased proliferation.

### 4.3. Sample Prep

Cells density was determined immediately before sampling by counting in Neubauer chamber. For every sample, 3 × 10^6^ cells were harvested. Cells were separated from the culture medium by centrifugation at 500× *g* for 4 min at 4 °C. An aliquot of the supernatant (“medium sample”) was frozen at −80 °C, and the rest of the supernatant was discarded. The cell pellet was briefly resuspended in 3% (*v*/*v*) glycerol in milliQ-water pre-cooled to 4 °C and centrifuged again at 500× *g* for 4 min at 4 °C. The wash solution was discarded and the cells were extracted with a modified Bligh-Dyer protocol [26] and resuspended in 1 mL 50:50 methanol:water pre-chilled to −20 °C. After the addition of 500 µL chloroform, samples were vortexed for 1 min and then centrifuged at 20,000× *g* for 1 min at 4 °C. Of the polar phase, aliquots of 120 µL were transferred to fresh 1.5 mL Eppendorf tubes. Of the organic phase, aliquots of 150 µL were transferred to 2 mL glass vials. All aliquots were dried in a vacuum centrifuge (Genevac EZ2, LMS Consult GmbH, Brigachtal, Germany) and stored at −80 °C until analysis.

### 4.4. FIA

The FIA method has been adapted from a previously published method by Fuhrer et al. [27]. Just prior to analysis, samples were resuspended in 150 µL FIA buffer. FIA buffer was 0.5% Agilent ESI-L Low concentration tuning mix in 80:20 acetonitrile:water. Subsequently, samples were diluted 10-fold in FIA buffer, transferred to sample vials with conical glass insert, capped, and placed in the autosampler. An aliquot of FIA buffer was used as a solvent blank sample.

An Agilent 1290 Infinity II UHPLC system was used to deliver samples to the mass spectrometer. A constant flow of 150 µL/min of carrier buffer was used to transport samples from the autosampler to the MS. Carrier buffer was for negative mode 10 mM NH4OH in 80:20 acetonitrile:water and for positive mode 10 mM ammonium formate in 80:20 acetonitrile:water. The injection volume was 3 µL and samples were injected twice (technical duplicates). Autosampler temperature was 5 °C.

The LC was coupled to a Bruker impact II QTOF MS (resolution > 50,000 FSR) equipped with ionBooster ESI source. The ESI source parameters were: End Plate Offset: 400 V, Capillary Voltage: 1000 V, Charging Voltage: 300 V, Nebulizer Pressure 4.1 bar, Dry Gas 3 L/min, Dry Gas Temperature 200 °C, Vaporizer Temperature 350 °C, Sheath Gas 240 L/h.

The mass spectrometer was operated in MS1 full scan mode with the following parameters: scan range 20 to 1050 *m*/*z*, Funnel 1 RF 200 Vpp, Funnel 2 RF, 200 Vpp, isCID 0 eV, Hexapole RF 50 Vpp, Quadupole Ion Energy 5 eV, Quadrupole Low mass 50 *m*/*z*, Collision Energy 2 eV, Pre Pulse Storage 5 µs, Scan rate 2 Hz. The mass axis was calibrated at the beginning of every batch of samples.

Samples were measured in a randomized order. Data were saved in profile mode and converted to mzML format using proteowizard/msconvert [28]. Subsequent data processing was performed using an in-house R script. Briefly, signals of the tune mix components were used for a quadratic re-calibration of the mass axis and for normalizing signal intensities. In every sample, the most intense 1000 *m*/*z* features were detected and subsequently combined to a consensus feature list. In the case of missing values, these gaps were filled by extracting signal intensities from the original data at the exact *m*/*z* values. Isotope peaks were recognized as consecutive chains of features with a mass difference of 1.00335 Da and probable intensity ratios and collapsed into one feature. Features were annotated by matching their mass (assuming M+H or M-H ions) to the masses of metabolites in the human metabolome database (HMDB) [29].

### 4.5. LC-QTOF-HILIC

The HILIC method has been adapted from a previously published method by Bajad et al. [30]. Just prior to analysis, samples were resuspended in 20 µL 90:10 acetonitrile:water. Samples were mixed and then centrifuged for 1 min at 20,000× *g* and 4 °C. Fifteen microliters of the supernatant was transferred to a 96-well PCR plate. The plate was sealed and then placed in the autosampler.

Chromatographic separation was performed on an Agilent 1290 Infinity II UHPLC system using a Phenomenex Luna NH2 column (50 × 2 mm, 3 µm particles). Buffer A was 10 mM NH4OH in water, Buffer B was 5 mM ammonium carbonate in 90:10 Acetonitrile:water. The gradient profile was: 0 min, 100% B, 0.25 mL/min; 0.5 min, 100% B, 0.25 mL/min; 0.7 min, 100% B, 1 mL/min; 1.2 min, 100% B, 1 mL/min; 5.4 min, 30% B, 0.75 mL/min; 5.7 min, 10% B, 0.75 mL/min; 7.7 min, 10% B, 0.75 mL/min, 8.3 min, 100% B, 0.75 mL/min; 8.6 min, 100% B, 1 mL/min; stop time: 9 min. The initial 0.5 min of each run were used for calibrating the mass axis. Then 3 µL of the sample was injected 0.7 min after the start of the run. The column temperature was 30 °C and the autosampler temperature was 5 °C.

The LC was coupled to a Bruker impact II QTOF MS (resolution > 50,000 FSR) equipped with ionBooster ESI source. The ESI source parameters were: End Plate Offset: 400V, Capillary Voltage: 1000 V, Charging Voltage: 300 V, Nebulizer Pressure 5 bar, Dry Gas 4 L/min, Dry Gas Temperature 250 °C, Vaporizer Temperature 400 °C, Sheath Gas 240 L/h.

The mass spectrometer was operated in negative mode with Auto MS/MS with the following parameters: scan range 20 to 1000 *m*/*z*, Funnel 1 RF 200 Vpp, Funnel 2 RF, 200 Vpp, isCID 0 eV, Hexapole RF 50 Vpp, Quadupole Ion Energy 5 eV, Quadrupole Low mass 50 *m*/*z*, Collision Energy 2 eV, Pre Pulse Storage 5 µs, Scan rate 12 Hz. Fragmentation was ramped from Collision RF 200 Vpp, Transfer Time 20 µs, Collision Energy 20 to Collision RF 700 Vpp, Transfer Time 70 µs, Collision Energy 50. The mass axis was calibrated at the beginning of every sample run.

Samples were measured in a randomized order. Data were acquired in Profile and Centroid Mode using Bruker Hystar 4.1 and processed using Bruker Metaboscape 5.

### 4.6. LC-QTOF-polarRP

Reversed-phase liquid chromatography has been widely used in metabolomics; this method has been adapted from previous methods [31,32,33,34]. Just prior to analysis, samples were resuspended in 20 µL water. Samples were mixed and then centrifuged for 1 min at 20,000× *g* and 4 °C. Fifteen microliters of the supernatant was transferred to a 96-well PCR plate. The plate was sealed and then placed in the autosampler.

Chromatographic separation was performed on an Agilent 1290 infinity II UHPLC system using a Waters CSH C18 column (100 × 2 mm, 1.7 µm particles). Buffer A was 0.1% formic acid in water, Buffer B was 50:50 acetonitrile:methanol. The gradient profile was: 0 min, 0% B, 0.2 mL/min; 0.5 min, 0% B, 0.2 mL/min; 0.6 min, 0% B, 0.4 mL/min; 4.7 min, 0% B, 0.4 mL/min; 19.7 min, 97% B, 0.4 mL/min; 25.2 min, 97% B, 0.4 mL/min; 25.7 min, 0% B, 0.4 mL/min, 27.2 min, 0% B, 0.45 mL/min; stop time: 28 min. The initial 0.5 min of each run were used for calibrating the mass axis. Then, 3 µL of sample was injected 0.7 min after the start of the run. Column temperature was 30 °C and autosampler temperature was 5 °C.

The LC was coupled to a Bruker impact II QTOF MS (resolution > 50,000 FSR) equipped with ionBooster ESI source. The ESI source parameters were: End Plate Offset: 400 V, Capillary Voltage: 1000 V, Charging Voltage: 300 V, Nebulizer Pressure 4.1 bar, Dry Gas 3 L/min, Dry Gas Temperature 200 °C, Vaporizer Temperature 350 °C, Sheath Gas 240 L/h.

The mass spectrometer was operated in positive mode with Auto MS/MS with the following parameters: scan range 20 to 1000 *m*/*z*, Funnel 1 RF 200 Vpp, Funnel 2 RF, 200 Vpp, isCID 0 eV, Hexapole RF 50 Vpp, Quadupole Ion Energy 5 eV, Quadrupole Low mass 50 *m*/*z*, Collision Energy 2 eV, Pre Pulse Storage 5 µs, Scan rate 12 Hz. Fragmentation was ramped from Collision RF 200 Vpp, Transfer Time 20 µs, Collision Energy 20 to Collision RF 700 Vpp, Transfer Time 70 µs, Collision Energy 50. The mass axis was calibrated at the beginning of every sample run.

Samples were measured in a randomized order. Data were acquired in Profile and Centroid Mode using Bruker Hystar 4.1 and processed using Bruker Metaboscape 5.

### 4.7. LC-QTOF-Lipid

This method has been adapted from previously published methods [35,36]. Just prior to analysis, samples were resuspended in 40 µL 2:1:1 2-propanol:acetonitrile:water and transferred to a sample vial with a conical glass insert. The vials were capped and then placed in the autosampler.

Chromatographic separation was performed on an Agilent 1290 infinity II UHPLC system using an Agilent Zorbax Eclipse Plus C18 column (100 × 2 mm, 1.8 µm particles). Buffer A was 10 mM ammonium formate in 60:40 acetonitrile:water. Buffer B was 10 mM ammonium formate in 90:10 2-propanol:acetonitrile. The gradient profile was: 0 min, 30% B, 0.2 mL/min; 0.5 min, 30% B, 0.25 mL/min; 0.6 min, 30% B, 0.4 mL/min; 1.2 min, 30% B, 0.4 mL/min; 5.2 min, 68% B, 0.4 mL/min; 21.2 min, 75% B, 0.4 mL/min; 21.7 min, 97% B, 0.4 mL/min; 26.0 min, 97% B, 0.4 mL/min; 26.5 min, 30% B, 0.4 mL/min; stop time: 28 min. The initial 0.5 min of each run were used for calibrating the mass axis. Then, 2 µL of sample were injected 0.7 min after the start of the run. Column temperature was 35 °C and autosampler temperature was 5 °C.

The LC was coupled to a Bruker impact II QTOF MS (resolution > 50,000 FSR) equipped with ionBooster ESI source. The ESI source parameters were: End Plate Offset: 400 V, Capillary Voltage: 1000 V, Charging Voltage: 300 V, Nebulizer Pressure 4.1 bar, Dry Gas 3 L/min, Dry Gas Temperature 200 °C, Vaporizer Temperature 350 °C, Sheath Gas 240 L/h.

The mass spectrometer was operated in negative or positive mode with Auto MS/MS with the following parameters: scan range 50 to 1600 *m*/*z*, Funnel 1 RF 200 Vpp, Funnel 2 RF, 200 Vpp, isCID 0 eV, Hexapole RF 50 Vpp, Quadupole Ion Energy 5 eV, Quadrupole Low mass 90 *m*/*z*, Collision Energy 5 eV, Pre Pulse Storage 5 µs, Scan rate 12 Hz. Fragmentation was ramped from Collision RF 400 Vpp, Transfer Time 50 µs, Collision Energy 20 to Collision RF 1200 Vpp, Transfer Time 100 µs, Collision Energy 50. The mass axis was calibrated at the beginning of every sample run.

Samples were measured in a randomized order. Data were acquired in Profile and Centroid Mode using Bruker Hystar 4.1 and processed using Bruker Metaboscape 5.

### 4.8. LC-MS/MS Data Processing

All LC-MS/MS data were processed using Metaboscape 5 (Bruker) with the parameters listed in Table 3:

Batch annotation of features was performed in three ways using Metaboscape: prediction of most likely sum formula using the SmartFormula function, annotation by matching retention time, exact mass, and isotope pattern to a list of known compounds using the AnalyteList function, and annotation by matching of exact mass and fragment pattern to spectral libraries. Chosen tolerances and other settings are listed in Table 4. We have removed stereochemical information (such as D- or L-) from annotated metabolite names after export from Metaboscape because the applied methodology cannot discriminate between stereoisomers.

### 4.9. Statistical Analysis

Feature tables (bucket lists) were exported from Metaboscape or generated by the FIA data processing pipeline described above. Consequently, data sets were quantile normalized (R function normalize.quantiles() from package preprocessCore [37]) to compensate for changes in cell size during the course of the experiment.

Dimensionality reduction for plotting was achieved by principal component analysis (PCA). To this end, quantile normalized data were square-root scaled to limit the impact of features with high absolute intensity. PCA scores and loadings were calculated using the R function princomp() from package core package “stats”.

Dimensionality reduction for plotting with simultaneous analysis identification of time-dependent features was achieved by partial least squares (PLS) analysis using sampling time as a predicted variable. The combined positive mode and negative mode FIA data set was log scaled and PLS was calculated using function plsr() from package pls [38].

The number of features that show a signal intensity above blank was determined by performing t-tests (one-tailed, equal variance) between a sterile well control (blank) and cell extracts separately for each time point, feature, and data set. A feature was defined to be above blank, if the quantile normalized signal intensity for at least one time point was greater than blank with a *p*-value < 0.05.

The number of features that exhibit a significant time trend was determined by performing a likelihood ratio test (R function lrtest() from package lmtest [39]) with a generalized additive model (R function gam() from package mgcv [40]. All features with a *p*-value < 0.05 (Bonferroni corrected for the number of features in the data set) were labeled as having a significant time trend.

### 4.10. Flow Cytometry

Every 24 h from 0 h to 72 h post T cell activation, cells were stained with the LIVE/DEAD Fixable Violet Dead Cell Stain Kit (Thermo scientific, https://www.thermofisher.com) and CD8a-PE (Biolegend, San Diego, USA, 100709) in 2% FBS/PBS on ice for 30 min. Cells were collected on a Fortessa flow cytometer (BD Bioscience, Haryana, India) and analyzed using FlowJo software (Version 10.1). The mean forward scatter (FSC-A) of CD8+, live cells was gated as a measure of cell size during T cell activation.

### 4.11. Graphical Abstract

The graphical abstract was prepared using elements generated with iPath3 [41].

## Figures and Tables

**Figure 1 metabolites-11-00012-f001:**
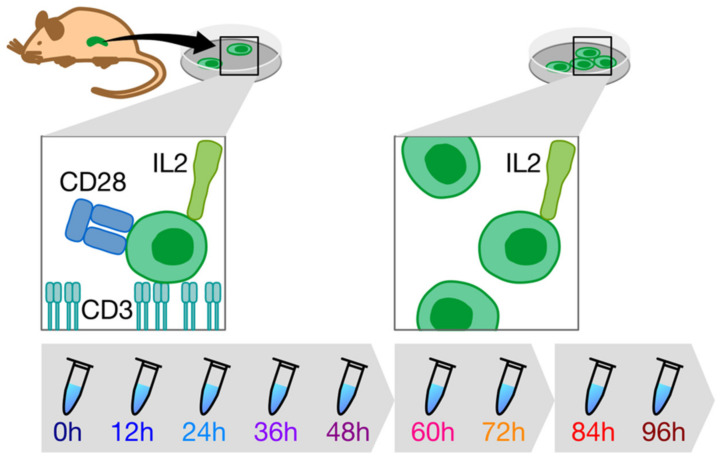
Schematic representation of the experimental design. Primary CD8+ T cells were isolated from murine spleens and cultured in vitro in the presence of activating stimuli CD3, CD28, and IL2. After 48 h and 72 h, the culture medium was exchanged for a medium containing survival growth factor IL2. Samples were taken every 12 h.

**Figure 2 metabolites-11-00012-f002:**
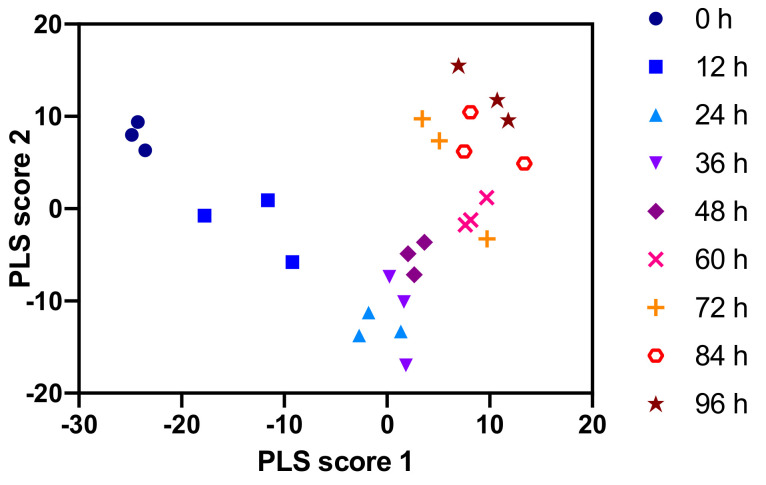
Partial least squares (PLS) analysis of a combined positive and negative mode FIA data set with time as a predicted variable. Groups of replicates are clearly separated for early time points but less well separated for late time points.

**Figure 3 metabolites-11-00012-f003:**
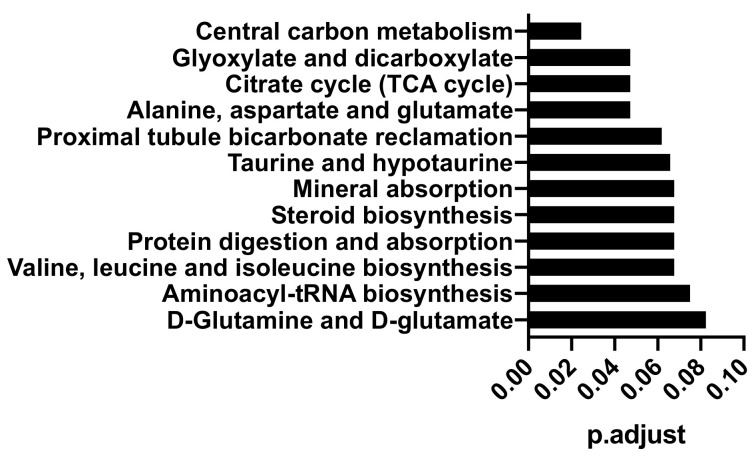
Top scoring KEGG metabolic pathways from gene set enrichment analysis (GSEA) using the sum of PLS scores of each measured feature as input.

**Figure 4 metabolites-11-00012-f004:**
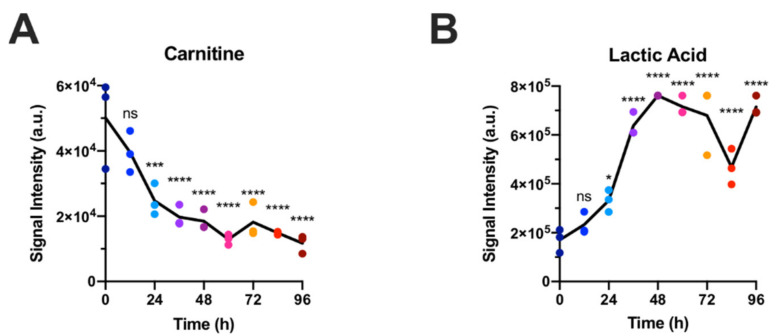
Time course of indicator metabolites for fatty acid oxidation ((**A**): carnitine) and glycolysis ((**B**): lactate) in LC-MS/MS data confirm previous findings that during activation, CD8+ T cells reduce fatty acid oxidation and increase glycolysis. Each panel represents one data set but data points have been colored by sampling time. Statistical significance was calculated by one-way ANOVA using GraphPad Prism Software comparing 3 biological replicates at each time point with the 0 h time point. ns = not significant, * *p* ≤ 0.05, *** *p* ≤ 0.001, **** *p* ≤ 0.0001.

**Figure 5 metabolites-11-00012-f005:**
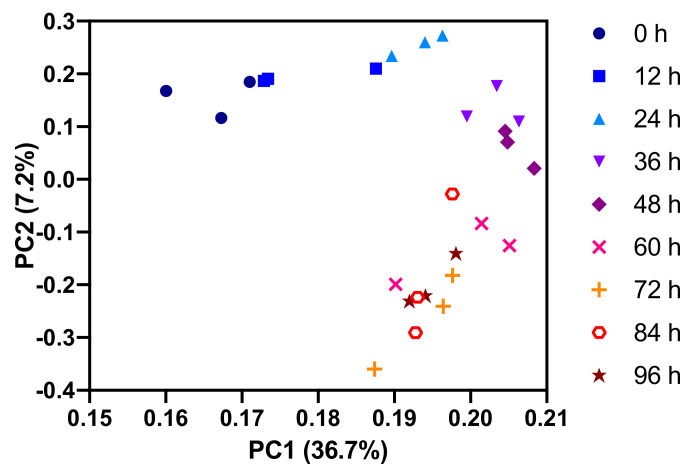
Principal component analysis of combined HILIC-negative mode MS and reversed-phase-positive mode MS data of the polar metabolome. Principal Component 1 (PC1) better separates the earlier time points, whereas PC2 better separates the later time points.

**Figure 6 metabolites-11-00012-f006:**
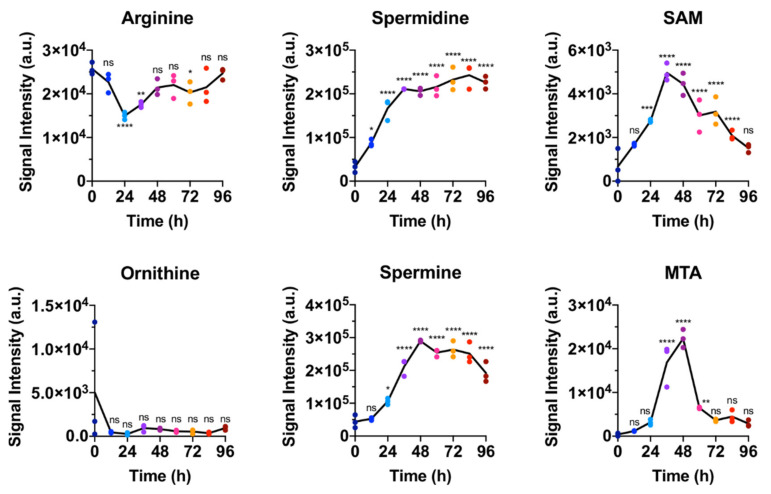
Time courses of key metabolites in polyamine biosynthesis measured by LC-MS show almost constant levels of the precursors arginine and ornithine, a persistent increase in spermidine and spermine, and transient increase in cofactors S-adenosyl-methionine (SAM) and methyl-thioadenosine (MTA). Each panel represents one data set but data points have been colored by sampling time. Statistical significance was calculated by one-way ANOVA using GraphPad Prism Software comparing 3 biological replicates at each time point with the 0 h time point. ns = not significant, * *p* ≤ 0.05, ** *p* ≤ 0.01, *** *p* ≤ 0.001, **** *p* ≤ 0.0001.

**Figure 7 metabolites-11-00012-f007:**
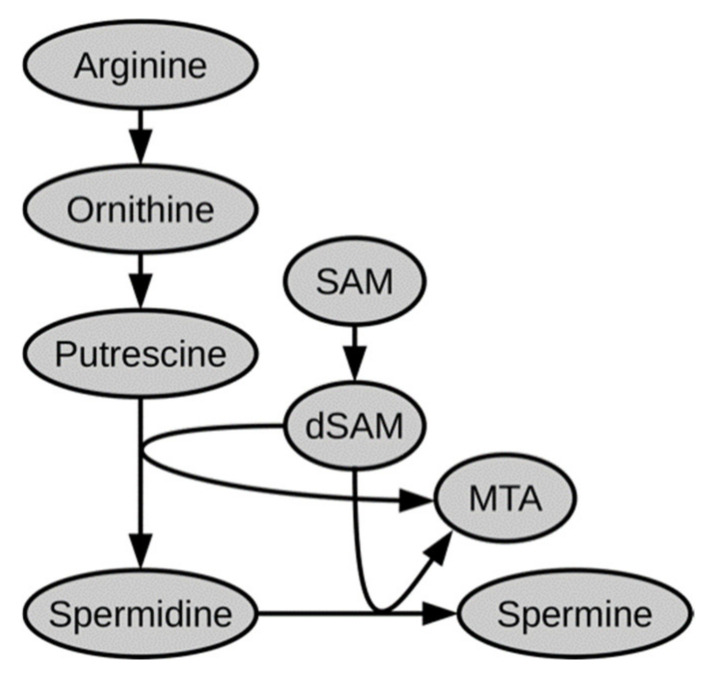
Schematic representation of polyamine biosynthesis pathway including cofactors s-adenosyl methionine (SAM), decarboxy-s-adenosyl methionine (dSAM), and methyl-thioadenosine (MTA).

**Figure 8 metabolites-11-00012-f008:**
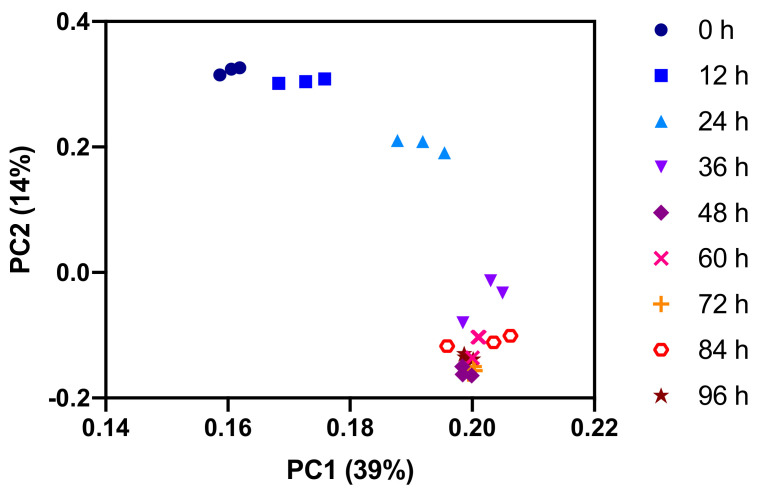
Principal component analysis of combined positive and negative mode LC-MS lipid data set. Only replicates of data points in the first half of the experiment are well separated.

**Figure 9 metabolites-11-00012-f009:**
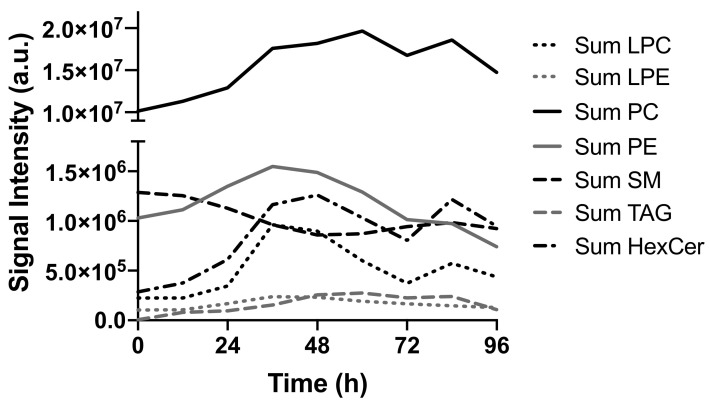
Summed signal intensity (not normalized for cell size effects) for major lipid classes show highly class-specific time trends. LPC: lysophosphatidylcholine, LPE: lysophosphatidylethanolamine, PC: phosphatidylcholine, PE: phosphatidylethanolamines, SM: sphingomyelin, TAG: triacylglycerol, HexCer: hexosylceramide.

**Figure 10 metabolites-11-00012-f010:**
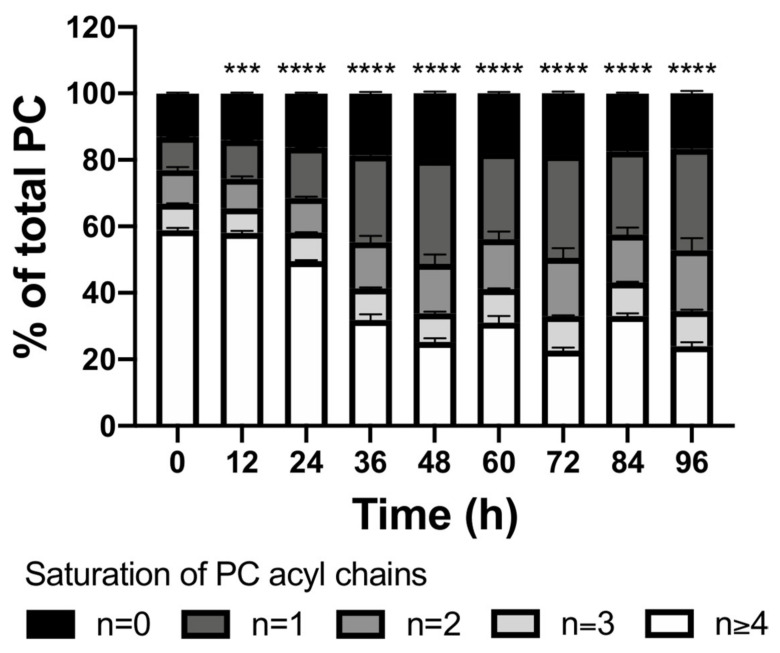
Saturation of PC acyl chain length over time displays a strong decrease of species with 4 or more double bonds. Statistical significance was calculated by one-way ANOVA using GraphPad Prism Software comparing 3 biological replicates at each time point with the 0 h time point. ns = not significant, *** *p* ≤ 0.001, **** *p* ≤ 0.0001.

**Figure 11 metabolites-11-00012-f011:**
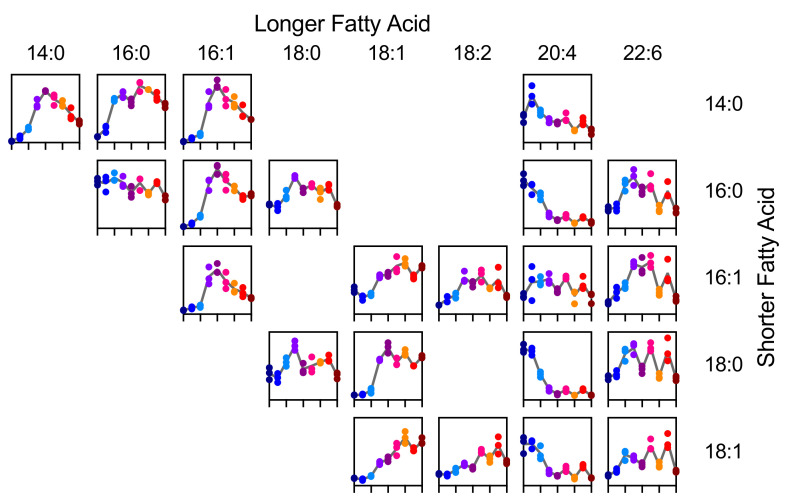
Time course of selected species of PC with a single fatty acid resolution with time 0 to 96 h on the *x*-axis and quantile normalized signal intensity on the *y*-axis. Each panel represents one data set but data points have been colored by sampling time.

**Table 1 metabolites-11-00012-t001:** Overview of analytical methods that were used for the broad coverage of metabolites. Abbreviations are as follows: FIA: flow injection analysis, MS1: one-dimensional mass spectrometry, MS2: two-dimensional mass spectrometry with the fragmentation of selected ions and measurement of fragment spectra, +: positive ionization mode, -: negative ionization mode, LC: liquid chromatography, QTOF: quadrupole-time-of-flight mass spectrometer, HILIC: hydrophilic interaction liquid chromatography, RP: reversed phase.

Method Name	Chromatography/Stationary Phase	Coverage	Elution Order	Detection	Polarity
FIA	none	All compounds that ionize well	No retention	MS1	+ and −
LC-QTOF HILIC	HILIC/aminopropyl	Amino acids, nucleotides, sugar phosphates, soluble cofactors, organic acids	Hydrophilic moieties: increase retention, hydrophobic moieties: little influence	MS1 and MS2	-
LC-QTOF polar RP	Reversed phase/C18	Acyl carnitines, nucleosides, nucleobases, some cofactors, some amino acids	Hydrophilic moieties: reduce retention, hydrophobic moieties: increase retention	MS1 and MS2	+
LC-QTOF Lipids	Reversed phase/C18	Glycerolipids, glycerophospholipids, sterols, sphingolipids	Hydrophilic moieties: reduce retention, hydrophobic moieties: increase retention	MS1 and MS2	+ and −

**Table 2 metabolites-11-00012-t002:** Overview of data sets generated for this study. Three different chromatographic methods and FIA were used. Most of the features that were detected above blank levels also exhibited a time-dependent trend in a likelihood ratio test using spline models (see methods section for details).

Method Name	Polarity	# Features	% Features above Blank	% Features above Blank and with Time Trend
FIA	-	1887	44.4	31.0
FIA	+	2416	52.7	28.4
LC-QTOF HILIC	-	1671	31.2	26.3
LC-QTOF Polar RP	+	1549	32.8	25.8
LC-QTOF Lipids	-	1745	57.9	53.4
LC-QTOF Lipids	+	1819	57.6	56.6

**Table 3 metabolites-11-00012-t003:** Parameters for data processing including batch feature annotation in Metaboscape 5.

Filter Parameters	
Minimum # Features for Extraction	1
Presence of features in minimum # of analyses	3
**T-ReX 3D Processing Parameters**	
Intensity threshold	4000 (polar metabolites), 3000 (lipids neg), 6000 (lipids pos)
Minimum Peak Length	12
Enable Recursive Feature Extraction	true
Minimum Peak Length (recursive)	7
Perform MS/MS import	true
MS/MS import method	average
**Ion Deconvolution Parameters**	
EIC correlation	0.8
Primary ion (negative mode)	[M-H]-
Primary ion (positive mode)	[M+H]+
Seed ions (negative mode)	[M+Cl]-
Seed ions (positive mode)	[M+Na]+, [M+K]+, [M+NH_4_]+
Common ions (negative mode)	[M-H-H2O]-, [M+COOH]-
Common ions (positive mode)	[M-H-H_2_O]+
**Mass Calibration Parameters**	
Lock Mass Calibration	false
Mass Recalibration	true, calibration segment 0.1-0.4 min
**Expert settings**	
FerraWorkflow.chargeMax	1 (only for polar metabolites)

**Table 4 metabolites-11-00012-t004:** Settings and parameters for feature annotation in Metaboscape.

Smart Formula Parameters	
*m/z* tolerance	1 mDa (narrow), 3 mDa (wide)
mSigma	15 (narrow), 50 (wide)
Elements	CHNOPS
Upper formula	S1
Element ratio filters	Common
Electron configuration	Both
**Analyte List Parameters**	
*m/z* tolerance	1 mDa (narrow), 3 mDa (wide)
Retention time tolerance	0.2 min (narrow), 0.4 min (wide)
mSigma	15 (narrow), 50 (wide)
**Spectral Library Parameters**	
Libraries (polar metabolites)	In house library, Bruker MetaboBASE 3.0, GNPS export (downloaded July 2020)
Libraries (lipids)	In house library, Bruker MetaboBASE 3.0, MSDIAL LipidDB VS68 (neg and pos)
*m/z* tolerance	1 mDa (narrow), 3 mDa (wide)
mSigma	20 (narrow), 200 (wide)
MS/MS score	900 (narrow), 700 (wide)

## Data Availability

Metabolomics data have been deposited to the EMBL-EBI MetaboLights database [42] with the identifier MTBLS2145. The complete dataset can be accessed here (as soon as the curation process by the MetaboLights team is completed): https://www.ebi.ac.uk/metabolights/MTBLS2145.

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
