# Peer review of "Metabolic Dynamics of In Vitro CD8+ T Cell Activation"

_metabolites, 2020, doi:10.3390/metabo11010012_

Round 1

Reviewer 1 Report

The manuscript entitled ”Metabolic dynamics of in vitro CD8+ T cell activation” describes longitudinal changes in metabolic programs of activated CD8+ T cells. Methodologies for in-depth parallel assessment of metabolomic profiles of CD8+ T cells.

Importantly, several technologies applied in the study allow for broad coverage of the metabolome. Manuscript presents important finding in the field of metabolomics. Authors also share huge dataset with other scientists, Broad time-course of the experiments, which will contribute to better understanding of metabolic patterns of CD8+ activation.

The manuscript is understandably written and present interesting and valuable data, which are openly shared for further investigations by other groups. the conclusions consistent with the evidence and arguments presented.To assure highest quality of the data few comments should be taken into consideration:

Major comments:

  1. Would it be possible to perform longitudinal biostatistical analysis of selected metabolites?
  2. Graphical abstract should be improved to better represent described findings.

  3. Major grammatical corrections can be applied.

Reviewer 2 Report

The manuscript entitled "Metabolic dynamics of in vitro CD8+ T cell activation" provides interesting information about changes occuring in cell culture over time after their activation. In my opinion, the following comments should be addressed before publication:

  • In the abstract, there are several acronyms that should be defined.
  • In the abstract, it isn't clear where the number of 11000 features comes from. Those are too many features, and based on the paper content it seems to be a mistake.
  • In Table 1, acronyms should be listed.
  • In my opinion, information in table 1 can be biased. For example, as a comment for FIA it says "fast". However, issues related to ionization suppression or enhancement are not included. For lipids, in the comment it is mentioned that it is sensitive to strong organic solvent; however, it is not clear what the authors mean. Is it related to chromatographic peaks shapes?? I suggest to delete the "comment" column, or improve the information provided. In the same table, it is listed just MS, and I believe the authors mean high resolution MS. A single quad instrument is also "MS" and it can't provide exact masses for any ion.
  • In table 2, it is not clear when you state "with time trend", does it mean that they had statistical differences at different times points? please include information about statistics in the Table's caption.
  • In Figure 3, information about what strategy was used to identify the compounds assigned to different pathways is missing. How did you identify the changing features? what strategy did you use to verify their identity?
  • In Figure 4, did you use any internal standard for these plots? what if there are interferences that vary at different sampling times and cause suppression of the response of carnitine, and enhancement of lactic acid. This is a possibility considering the lack of chromatography. 
  • Figure 5: Does this figure contains positive and negative data? Did you curate the features to make sure that you were not plotting twice the same information (some metabolites can be retained under both reversed phase and HILIC conditions).
  • Figure 6: what about carnitine and lactic acid? did you observe the same changes in the LC-MS data? were you able to confirm your observations via FIA? a comment on the reliability of FIA based on LC-MS findings is missing.
  • In section 4.2: can you describe more details about how was the sampling/collection?
  • In materials and methods: were the LC-MS methods developed for this work, or did you follow conditions published somewhere else? if these are methods previously reported, please make sure to include appropriate citations.
  • Detailed information about features/metabolites identification is missing. Did you verify their identity via data bases matching only? did you run standards in any of the cases?

Reviewer 3 Report

it is well-written and present very interesting results T-cell metabolomic changes over time during activation. It is certainly of interest for Metabolites audience. The methodologies used are robust.

My only concern is regarding the statistical analyses. To my knowledge Partial Least Square (PLS) and Principal component analysis (PCA) should not be used with time-dependent datasets. Maybe the authors should justify why they used these approaches and how this could affect their biological interpretation.

Reviewer 4 Report

Edwards-Hicks et al present a study looking at changes in metabolism of CD8+T cells over a 96-hour time course. More studies like this are needed to provide researchers interested in immunometabolism a bench mark for metabolic time courses, both for small molecule metabolites and lipids.  While this study is highly relevant to the field a more detailed understanding of the analytical process and metabolite ID determination is warranted.  It is not possible to assess the analytical soundness of the data presented from the described currently described methods section.  

Major Comments:

Table 1: Technical definitions are not used to describe chromatographic separation methodologies and mass spectrometry scan types.  It would behoove the authors to use such terminology, inserted as another column or to replace what is written.   Definitions can be found in numerous analytical chemistry resources. Specific examples include the following:

FIA is not a type of chromatography as there is no separation and should not even be listed under chromatography.

Under HILIC comments—instead of using the word “sensitive” it would be technically correct to discuss the “elution” of metabolites is dependent based on buffer pH, etc,.   Elution is based on compound and solvent polarity, but the elution order is opposite that of reversed phase where less polar metabolites will elute first.  In reversed phase chromatography—elution is based on the hydrophobic interactions between metabolite and stationary phase or in other terms the polarity of the metabolites and chosen organic solvent system.

Reverse phase---Is technically Reversed phase chromatography

Lipid—lipid is not a chromatographic technique, but rather the metabolites being separated.  Lipids are also most commonly separated using a reversed phase column (e.g. C18 or C8); however, solvents differ from what is used for more hydrophobic small molecule metabolites as lipidomics analysis is focused on more hydrophobic species including PL and TGs.   

Under lipid “coverage” mono, di and tri could be condensed to glycerolipids. Cholesterol esters could be generalized to sterols and ceramide and sphingomyelin could be made broader and classified as sphingolipids.

Under Mass Spectrometry I would make the definitions for polarity more generic and just use what is written as examples of what may ionize better in one polarity vs the other.  It is also unclear what is meant by “polyunsaturated metabolites” under positive mode.  For example, fatty acids are polyunsaturated, but detection is driven by the carboxylic acid in negative ion mode unless derivatized or the pH of the solvent system is altered.

MS and MS/MS are scan types that mass analyzers can perform, not types of mass spectrometry.   What is written in the “comments” section are just ways they can be used in untargeted metabolomics, but not what function they perform.  This should be corrected to provide a definition for these scan types.

Line 49: This statement is speculative “Many metabolic studies are limited by single metabolite extraction methods (e.g. polar), single liquid-chromatography (LC) methods (e.g. HILIC), and single mass spectrometry (MS) polarities (e.g. positive mode), which do not account for the wide-ranging physiochemical properties within the metabolome, and therefore bias towards particular pathways (e.g. amino acid metabolism).”   In many cases analysis may be limited as the focus is limited, but there are also numerous examples of studies that have used multiple modes of separation and both polarities. In fact, if those who are not MS savvy choose to send samples to Metabolon, it is guaranteed their analysis will be done over multiple chromatographic modes in both ionization modes.  What I would say is more important about the current study is the longitudinal analyses that is being conducted under specific activation conditions. 

Feature vs Metabolite:  The words feature and metabolite are used interchangeably throughout the text; however, they were not defined for the non-MS savvy reader.  In addition, it is not understood as to whether the metabolites that are described in Figures 4 and 6 are confirmed identifications or putative identifications.  What if any features/putative IDs are confirmed based on either an in house library or follow up MS experiments?

Experimental Detail: Detail is lacking regarding quality control measures taken during sample prep and MS analysis. See below:

It is unclear what the cell lysate control is? If it is the blank cell media this is not correct, but rather the cell lysate extracts at each 12-hr time point should be compared to time 0.

Were internal standards used to determine extraction variability or to determine if the instrument stability and sensitivity changed over time? Was the internal mass cal used to normalize signal intensity as well as mass accuracy? What is the ISTD CV over each type of experiment? It appears that an ISTD was used for normalization in the supplemental data sheets, but no information is given regarding these steps and checks.

Were pooled QC samples run throughout the experiments?

To this point, Figure 4 reports lactate and carnitine signal intensity without normalization.  Peak area should be reported as a ratio to peak area of ISTD or peak area alone can be graphed if the ISTD peak area CV is low across the entirety of the experiment.

Line 127: Why is FIA unsuitable for lipids? Many people use FIA for untargeted lipidomics.

Line 170: A comparison should not be made between PCA for polar metabolites and lipids as a separation step was included in the lipid analysis thus more molecular ions could be detected as ion suppression was distributed over a third dimension, which was not the case for polar metabolites by FIA.

Line 257: Were changes in metabolism affected by cells being counted and replated every 24-hr?

What was the mass resolution for each scan type?

Data analysis: Based on the supplemental excel spreadsheets it is unclear if MS or MS/MS data (or both or a mix) was used for putative ID assignments.

Were any of these putative IDs confirmed?

Based on graphs it appears that there is an n = 3 for each time point. If only 3 samples per time point how can gap filling be used? Occupancy should be set at 100%. Are these technical or biological replicates?

There are no adducts listed in the excel spreadsheets for the data.  Many lipids, such as PL and TG, may be detected as M+H+ and M+NH4+ in positive mode and the aforementioned formate adducts in negative mode.  How was this data handled? Are all isotopes and adduct features collapsed into one “compound” or “metabolite” formula?

If only M-H+ or M+H+ is used to match to HMDB this would not take into account other adducts.

Why is signal intensity used and not peak area for comparison? Signal intensity is often not reflective of the relative amount if all peak widths are not consistent across the chromatographic separation.

Line 228: The discussion focuses on how the paper outlines methodologies, but these were not well described in the manuscript. Furthermore, it is unclear whether a large amount of the metabolome is covered if these are feature level or putative ID level rather than confirmed IDs.

Supplemental figures are mentioned, but were not in the supplemental file download with the excel tables.

Minor Comments:

Line 173:  hexosyl-ceramides

Line 176: sphingomyelin

Line 334: autosampler
